# A Framework for Optimal Navigation in Situations of Localization Uncertainty

**DOI:** 10.3390/s23167237

**Published:** 2023-08-17

**Authors:** Charifou Orou Mousse, Mohamed Benrabah, François Marmoiton, Alexis Wilhelm, Roland Chapuis

**Affiliations:** Université Clermont Auvergne, Centre National de Recherche Scientifique, Clermont Auvergne INP, Institut Pascal UMR6602, F-63000 Clermont-Ferrand, France; charifou.orou_mousse@uca.fr (C.O.M.); mohamed.ben_rabah@uca.fr (M.B.); francois.marmoiton@uca.fr (F.M.); alexis.wilhelm@uca.fr (A.W.)

**Keywords:** path following, obstacle avoidance, localization uncertainty

## Abstract

The basic functions of an autonomous vehicle typically involve navigating from one point to another in the world by following a reference path and analyzing the traversability along this path to avoid potential obstacles. What happens when the vehicle is subject to uncertainties in its localization? All its capabilities, whether path following or obstacle avoidance, are affected by this uncertainty, and stopping the vehicle becomes the safest solution. In this work, we propose a framework that optimally combines path following and obstacle avoidance while keeping these two objectives independent, ensuring that the limitations of one do not affect the other. Absolute localization uncertainty only has an impact on path following, and in no way affects obstacle avoidance, which is performed in the robot’s local reference frame. Therefore, it is possible to navigate with or without prior information, without being affected by position uncertainty during obstacle avoidance maneuvers. We conducted tests on an EZ10 shuttle in the PAVIN experimental platform to validate our approach. These experimental results show that our approach achieves satisfactory performance, making it a promising solution for collision-free navigation applications for mobile robots even when localization is not accurate.

## 1. Introduction

The plethora of existing works in the literature on path planning and obstacle avoidance [1,2,3,4,5] indicates that the issues surrounding autonomous navigation of mobile robots are far from being resolved and highlights the growing interest in addressing these problems. Most of the proposed approaches suffer from either realism (real-time execution is not feasible due to high computational times) or exhaustiveness (scenario-dependent applications that do not consider certain aspects), only partially addressing the problem. One of the major challenges that limits the proposed solutions is the vehicle localization estimation and the way to deal with its uncertainty. Localization is part of the primary level of autonomy (alongside environment mapping), ensuring the perception capability of a mobile robot in the “sense–plan–act” strategy [6]. Classical path planning and obstacle avoidance techniques generally assume that the robot is well localized before and while navigating. Thus, pioneering research in autonomous navigation initially focused on state estimation, aiming for accurate knowledge of the position and orientation of robots in the world. In [7,8], state estimation was recognized as the central problem for achieving autonomous navigation successfully. The more efficient proposed solutions at present involve probabilistic methods such as Bayesian filtering techniques. The Kalman filter [9] and its variants (UKF [10], CIF [11], etc.), along with particle filters [12], are widely used to merge data from multiple sensors (odometry, GNSS, lidars, cameras) to reduce uncertainty in robot localization. However, whatever the measures taken and techniques used, absolute localization remains fraught with uncertainty. Furthermore, for applications in rural and peri-urban areas we are never safe from situations of GNSS signal losses, for example when passing under tunnels or dense trees and in unfavorable weather conditions. Therefore, uncertainty should be explicitly taken into account during planning and navigation. There is no doubt that a robot which is poorly localized in the world will perform poorly in achieving an objective such as following a trajectory. However, whatever the error in its absolute localization, the ability to navigate in a local environment built up by the robot’s on-board sensors should not be compromised. Indeed, the obstacles used for detection are usually projected in the world reference. Avoiding them while following a path in the world makes it mandatory to add the robot’s localization uncertainty to these obstacles’ positions, which loosens the initial detection accuracy (see Section 2.2). In this work, we focus on demonstrating this point and describe how the proposed framework deals with it.

In this work, we propose a robust technique for managing uncertainty to improve the reliability and safety of autonomous vehicles, with the goal of minimizing situations in which stopping the vehicle due to localization uncertainty becomes necessary. This approach allows the vehicle to operate in a minimalist degraded mode for path-following while maintaining optimal and robust obstacle avoidance capabilities in such situations. Optimal, in this sense, means that the vehicle makes the minimum deviation from the reference path, passing as close as possible to obstacles without colliding with them.

Our contributions in this paper concern:A criterion for path following in the presence of uncertainty.An obstacle avoidance approach that is robust against localization uncertainty,A framework that optimally unifies path-following and obstacle avoidance by keeping each objective in the preferred frame of reference.Results from real experiments with a real autonomous shuttle for two different scenarios, namely, navigating in the absence and presence of localization uncertainties.
The rest of this paper is organized as follows. In Section 2, we introduce the state of the art in path planning and navigation, focusing on the Dynamic Window Approach and tentacles-based approaches. In addition, we review the research on navigation in the presence of localization uncertainty. In Section 3, we provide the main principles of our rigorous framework, explaining how it effectively integrates path following and obstacle avoidance capabilities while maintaining their independence. Section 4 is dedicated to experimental results for different scenarios with and without localization uncertainties implemented on an EZ10 robotized vehicle (see Section 4.1.2). We discuss these results. Finally, we summarize the research contributions of this paper in Section 5 and outline potential future research avenues.

## 2. Related Work

### 2.1. Planning and Navigation

A categorization that is commonly made in path planning approaches is to distinguish between local and global techniques. From another perspective, we can associate path planning with global approaches and navigation with local techniques. In fact, global approaches are typically performed using prior maps that represent the entire navigation space (cell decomposition, visibility graph, Voronoi diagram) [13] and attempt to generate a path from a starting point to a goal point while avoiding static known obstacles and optimizing other criteria such as distance traveled and energy consumption. When a path is found, it should be followed using a path-following control law such as Model Predictive Control, pure pursuit [14], or Samson’s law [15].

On the other hand, local navigation techniques are designed to be reactive to changes in the robot’s immediate environment and to take into account the possible presence of obstacles. The principle is to continuously analyze the traversability of the path in front of the robot and obtain a trajectory that optimizes a criterion minimizing the risk of collision. Thus, a dynamic representation of the environment needs to be derived from sensor data. The Bayesian occupancy grid [16] is the most common model used to evaluate the collision risk over a path. Depending on the application, however, a representation can be limited, prompting the use of other frameworks such as elevation maps [17], lambda fields [18], and knowledge maps [19]. Because the robot always has a global objective to achieve in the world, reactive approaches are in fact hybrids, allowing the outputs of global techniques to be taken into account. Several of the most prominent works in the state of the art have dealt with Artificial Potential Field [20], Dynamic Windows Approach (DWA) [21], and tentacles-based approaches [22].

#### 2.1.1. Dynamic Windows Approach

Dynamic Windows Approach (DWA) [23] consists of generating a set of trajectories at each iteration, taking into account the robot’s kinematic and dynamic constraints and evaluating these trajectories to select the best one. Admissible trajectories are obtained from a set of linear and angular velocity pairs (v,ω) which define the space of velocities attainable by the robot. They are circular because of the robot model, except in the exceptional case of zero angular velocity. The space of attainable speeds is defined by
(1)$$V_{r}=V_{s}\cap V_{a}\cap V_{d}$$

Here, $V_{s}$ is the set of possible speeds according to robot’s actuators specifications.
(2)$$V_{s}=\left\{\left(v,\omega \right)\;|\;v\in \left[v_{min},v_{max}\right]\wedge \omega \in \left[\omega_{min},\omega_{max}\right]\right\}$$

Let $\left({\dot{v}}_{b},{\dot{\omega }}_{b}\right)$ be the linear and angular accelerations and let $V_{a}$ represent the set of admissible speeds, enabling the robot to stop before colliding with known static obstacle. If the position of static obstacles in the environment is not known, this space is not definable and can be omitted. In order to take account of the limiting accelerations exerted by the motors, the velocity space is reduced to the dynamic window, which contains only the velocities instantaneously attainable at the next sampling period dt. Let $\left(v_{k},\omega_{k}\right)$ be the velocities of the robot at time *k* and $\left(\dot{v},\dot{\omega }\right)$ be the maximum accelerations applicable to the next sampling time. The set of instantly attainable speeds $V_{d}$ at time k+1 is defined by
(3)$$V_{d}=\left\{\left(v,\omega \right)\;|\;v\in \left[v_{k}-\dot{v}\cdot dt,v_{k}+\dot{v}\cdot dt\right]\wedge \omega \in \left[\omega_{k}-\dot{\omega }\cdot dt,\omega_{k}+\dot{\omega }\cdot dt\right]\right\}.$$

Here, dt is the time between *k* and k+1 discrete times. This space of commands $V_{r}$ is discretized and each couple (v,ω) defines a trajectory predicted with the robot model during a time horizon (see  Figure 1) and evaluated with three criteria to foster high speeds: target heading, clearance for obstacle avoidance, and velocity. Notice that this stage is an optimization process that allows for both nonlinearity and dynamic robot models to be taken into account, as well as the actuators’ potential delays.

#### 2.1.2. Tentacles-Based Approach

The main idea behind tentacle-based navigation is to simulate the behavior of arthropods such as insects, which have antennae on their heads used for olfaction, touch, and taste thanks to sensory structures called “sensilla” [24]. In the context of autonomous vehicle navigation, the principle is to generate virtual antennae called “tentacles” that extend into the robot’s environment to detect obstacles and find collision-free paths. These tentacles can be in the form of circular arcs, clothoids, or parallel to the road. This approach was implemented in 2007 during the C-ELROB (Civil European Land-Robot Trial) and the DARPA Urban Challenge on two vehicles with the objective of navigating in unstructured and unknown environments [22]. The immediate environment of the vehicle is perceived by a Velodyne 64-beam 360-degree Lidar and the measurements are converted into an egocentric Bayesian occupancy grid associated with the vehicle. The robot’s footprint on a tentacle is used to extract the probability of collision on that tentacle from the occupancy grid. Tentacles are then classified as either navigable (obstacle-free) or non-navigable.

Among the navigable tentacles, a combination of three criteria are used to choose the best one, taking into account the following:Tentacle clearance, i.e., the distance to the nearest obstacle (clearence)Smoothness of steering ($V_{smooth}$) as a function of the variation in tentacle curvatureConvergence towards a reference trajectory to follow ($V_{traj}$).

The criterion of convergence to a reference trajectory is the advantage offered by the tentacles method in comparison with DWA, in which modeling allows both obstacle avoidance and smooth steering. This path-following criterion minimizes the vehicle’s future lateral and angular deviation from the reference trajectory. For each tentacle, a distance measure $v_{dist}=a+c_{\alpha }\alpha$ is calculated with *a* and α being the future lateral and angular deviation, respectively, and $c_{\alpha }=0.3\;\mathrm{rad}/m$ being a constant that serves as a unit conversion factor from angular deviation α to linear deviation. The normalized final cost for each trajectory is defined as
(4)$$v_{traj}=\frac{v_{dist}-v_{min}}{v_{max}-v_{min}}.$$

Here, $v_{max}$ and $v_{min}$ are the maximum and minimum calculated values of $v_{dist}$. Indeed, many of these approaches and integrated criteria do not take into account the potential uncertainty in localization, and assume that the robot is precisely localized.

### 2.2. Navigation under Localization Uncertainty

Navigating with localization uncertainties is an active area of research, and the majority of studies on this subject propose modeling the problem as a Partially Observable Markov Decision Process (POMDP) [25,26,27]. Previously overlooked due to their high computational complexity, POMDP approaches have made tremendous advances since the early 2000s thanks to sampling-based approximate solvers [27]. In the case of autonomous vehicle navigation, these techniques investigate the influence of uncertainty propagated along the trajectory on the decisions made by planning algorithms. Due to the uncertainty, a POMDP agent never knows its exact state; therefore, it cannot decide the best action based on a single state. Instead, a POMDP agent decides its action based on a set of states that are consistent with the available information. It represents a set of possible states in the form of a distribution over the state space, called a belief. A POMDP agent plans in this belief space, which is the set of all possible beliefs, by considering several different states in this space. Therefore, the agent’s detections are uncertain, as they are associated with each of these possible states. When using this framework, uncertainty affects the robot’s global objectives (trajectory tracking, navigation to a target point) and additionally propagates onto locally detected obstacles.

In [28], a Gaussian propagation technique was used to propagate uncertainty along an occupancy grid. In this way, the uncertainty is taken into account when assessing risk, making the system more reliable in its motion planning process. This kind of technique was used in [19] to blur a prior map by a Gaussian distribution expressing the localization uncertainty. By proceeding in this manner, obstacles are dilated by convolution with the covariance matrix. To avoid such an uncertain obstacle, the robot should keep an inflated safety distance (Figure 2b).

However, if we consider a robot equipped with detection sensors and an obstacle avoidance module, and which has the aim of maintaining a heading when encountering an obstacle, whatever its absolute uncertainty, it will avoid the obstacle with the optimum safety distance. Indeed, propagation of the robot’s absolute uncertainty in its local detection is due to the fact that we want to leverage prior information, especially for following a reference trajectory in the world, and utilize local information for obstacle avoidance by bringing these two objectives into a common absolute reference frame.

Our strategy is to keep the vehicle detections in its local reference frame. These detections are accurate to within sensor errors, and enable the vehicle to avoid them in the vehicle-centric reference frame regardless of its absolute location in the world. However, the path-following objective, which is related to the absolute reference frame of the world, is affected by the accuracy of its absolute localization (Figure 2c).

## 3. Proposed Approach

### 3.1. Environement Model

In our approach, the robot’s workspace is represented by a robot-centered occupancy grid [29] generated by 2D Lidar sensors (other sensors, such as cameras, etc., can contribute as well). This map retains none of the previous data, and simply transforms the Lidar scans into a grid using a binary Bayesian filter. It should be noted that in our case we have opted to rotate the robot within the map instead of rotating the map itself, which is done to nullify the errors and computational costs caused by rotation (see  Figure 3). Another advantage of using such a map is to leverage the precision of Lidar sensors, which reliably detect obstacles; thus, there is no need to represent uncertainties linked to these obstacles, as they are not transformed to the global reference frame.

The occupancy grid used in our applications is 60 × 60 (m). Each cell is 20 × 20 (cm) square, meaning that the grid map is precisely a 299 × 299 matrix centered on the robot.

### 3.2. Path-Following Criterion

To define a trajectory-following criterion that takes into account localization uncertainty, let us consider a robot with state ${\left[x_{rw},y_{rw},\theta_{rw}\right]}^{⊤}$ and a covariance matrix $Q_{rw}$ in an absolute world reference frame. Using DWA, we generate a set of trajectories in the robot local frame (Figure 4). Let $P_{rl}={\left[x_{rl},y_{rl},\theta_{rl}\right]}^{⊤}$ be the last state in a predicted trajectory. The transformation of this point from the vehicle’s local frame to the absolute world frame is provided by the following equation:(5)$$P_{rw}=f\left(x_{rw},y_{rw},\theta_{rw}\right)=R.P_{rl}+\left[\begin{array}{c} x_{rw}\\ y_{rw}\\ \theta_{rw} \end{array}\right]\;\mathrm{with}\;R=Rot\left(z,\theta_{rw}\right)=\left[\begin{array}{ccc} \cos(\theta_{rw}) & -\sin(\theta_{rw}) & 0\\ \sin(\theta_{rw}) & \cos(\theta_{rw}) & 0\\ 0 & 0 & 1 \end{array}\right]$$

The Jacobian matrix associated with this transformation in terms of $x_{rw}$, $y_{rw}$, and $\theta_{rw}$ is
(6)$$J_{f}=\left[\begin{array}{ccc} \frac{\partial f}{\partial x_{rw}} & \frac{\partial f}{\partial y_{rw}} & \frac{\partial f}{\partial \theta_{rw}} \end{array}\right]=\left[\begin{array}{ccc} 1 & 0 & -x_{rl}\sin\left(\theta_{rw}\right)-y_{rl}\cos\left(\theta_{rw}\right)\\ 0 & 1 & x_{rl}\cos\left(\theta_{rw}\right)-y_{rl}\sin\left(\theta_{rw}\right)\\ 0 & 0 & 1 \end{array}\right]$$

Thus, the covariance associated with the predicted local position of the robot projected into the world frame $Q_{P_{rw}}$ is provided by
(7)$$Q_{P_{rw}}=J_{f}\;Q_{rw}\;J_{f}^{⊤}$$

Considering the uncertainty in the future position of the robot in the world, its future lateral deviation from the reference trajectory to be followed cannot be determined by the Euclidean distance alone. The distance measure that is well-suited in this case is the Mahalanobis distance. It is defined as the Euclidean distance between two points weighted by the inverse of the covariance matrix.

  Thus, the Mahalanobis distance $M_{dist}$ between the robot’s future position in the world $P_{rw}$ and its orthogonal projection $M={\left[x_{M},y_{M},\theta_{M}\right]}^{⊤}$ onto the reference trajectory is provided by
(8)$$M_{dist}=\sqrt{{\left(P_{rw}-M\right)}^{⊤}Q_{P_{rw}}^{-1}\left(P_{rw}-M\right)}$$

The final normalized cost function for each predicted path P is provided by
(9)$$\Phi \left(P\right)=\frac{M_{dist}-M_{min}}{M_{max}-M_{min}}$$
where $M_{max}$ and $M_{min}$ are the maximum and minimum calculated values of $M_{dist}$.

### 3.3. Obstacle Avoidance Criterion

Recall that locally predicted trajectories are accurate in the robot-centric map; the uncertainty propagated on these trajectories is only due to their transformation to the absolute reference frame, in which the robot could be inaccurately localized and must follow a reference.

For each trajectory, we define the footprint of the robot and extract the path occupancy. The technique involves moving a bounding box of the vehicle shape at each predicted state, forming a polygon that is subsequently rasterized to produce a mask for extracting the path occupancy (see  Figure 5). In our case, we have chosen to represent the vehicle by an ellipse, where the major diameter is the robot’s length and the minor diameter is its width. While a rectangular shape would provide a more faithful representation and work equally well as an ellipse, it consumes more computational time.

After the occupancy probability has been extracted for a path, the first step is to classify it as navigable or not [22]. Thus, we eliminate trajectories that contain at least a certain number λ of occupied cells (determined according to the size of the grid cells and the nature of the potential obstacles that might be encountered).

Because a navigable trajectory is not necessarily risk-free, we need to define a risk cost function to choose the least risky trajectory. This risk of collision is defined as the integral over the path of the kinetic energy of contact weighted by the probability of occupancy [19]. We can then express our risk function Ψ(P) along a path P by
(10)$$\Psi \left(P\right)=\int_{P}\frac{1}{2}\cdot m\cdot v{\left(s\right)}^{2}\cdot P\left(s\right)ds$$
where *m* is the robot’s mass and v(s) and P(s) are respectively the instantaneous speed of the robot and the probability of occupancy at the abscissa *s*.

Because the occupancy grid saves a tessellated environment in which each cell stores the probability of occupancy of the underlying space, and because the robot’s mass and velocity for one and same trajectory are both constant, we can write
(11)$$\Psi \left(P\right)=\frac{1}{2}\cdot m\cdot v{\left(t_{j}\right)}^{2}\sum_{i=1}^{N}P\left(c_{i}\right)$$
where $t_{j}$ represents the trajectory to be evaluated, *N* is the total number of cells crossed by the path P, and $P(c_{i})$ is the probability of occupancy of the *i*th cell.

### 3.4. Global Cost

As shown in  Figure 1, different linear velocities can be considered for a direction given by an angular velocity or a steering angle, resulting in the same circular arc with a longer or shorter distance depending on the velocity. Furthermore, obstacles in the closer environment of the robot impose restrictions on the angular and linear velocities. For example, the maximal admissible speed on a trajectory depends on the distance to the next obstacle on this trajectory. For this reason, we have added a third term to our objective function to enable the robot to choose a suitable velocity from the set of velocities available in the dynamic window.

Therefore, the objective function to be mimimized for the retained trajectories is
(12)$$C\left(P\right)=\alpha \cdot \Phi \left(P\right)+\beta \cdot \Psi \left(P\right)+\gamma \cdot \Theta (v\left(t_{j}\right))$$
where α, β, and γ are weights to be tuned and $\Theta (v\left(t_{j}\right))$ is provided by
(13)$$\Theta \left(v\left(t_{j}\right)\right)=\frac{v_{max}-v\left(t_{j}\right)}{v_{max}}$$
with $v_{max}$ being the vehicle’s maximum admissible velocity.

## 4. Experiments and Results

In this section, we present experiments that validate the presented framework. First, the experimental setup for the tests is shown. Then, results are presented for two different scenarios: navigation with perfect localization, and navigation with biased localization. Finally, we discuss the obtained results.

### 4.1. Setup

#### 4.1.1. PAVIN Plateform

PAVIN (Plateforme Auvergnate pour les Véhicules INtelligents) is a 5300 m${}^{2}$ experimental site dedicated to the study of autonomous vehicle mobility in urban and natural environments (see  Figure 6). It provides a realistic simulation of a city center with urban backdrops and traffic lanes as well as a natural environment. We conducted our experiments on this platform.

#### 4.1.2. EZ10

Here, we describe the EZ10-type vehicle used to validate the framework. The EZ10 shuttle (see  Figure 7) is an electric vehicle designed by the Ligier Company© for automatic passenger transport. It is the result of several collaborative projects aimed at facilitating the introduction of autonomous vehicles in industrial and urban areas.

 Table 1 shows the technical specifications of the EZ10.

The vehicle is equipped with a number of sensors used in our applications. Proprioception is ensured by three odometric sensors which return the front and rear steering angles at 20 Hz as well as the forward speed measured on the engine shaft. The EZ10 features two cameras (front and rear) used for visual odometry [30] (only used here as a ground truth when combined with GNSS). A Kalman filter is used to merge these odometric data with data from an RTK-GPS (Real Time Kinematic Global Positioning System), which provides centimeter localization reference truth.

Planar lidars are mounted on all four corners of the vehicle to provide a $360^{\circ }$ scan of the entire vicinity over a distance of 50 m (lidars maximum range). Each lidar has an angular range of $-135^{\circ }$ to $135^{\circ }$. Lidar data are provided at 20 Hz. Only the two front lidars were used to build our occupancy grid, as we did not plan to drive in reverse. The scans are merged in a common frame related to the vehicle using the ROS [31] ira-lasers tools package [32].

The on-board computer runs Linux with ROS 1 used as middleware to manage the software bricks. All our experiments were carried out under ROS with Python and C++ nodes.

We approximate the vehicle model with an Ackermann model with two control parameters, namely, the steering angle δ and velocity *v*:(14)$$\left\{\begin{array}{c} \theta_{k}=\theta_{k-1}+\frac{v}{L}\tan\left(\delta \right)dt\\ x_{k}=x_{k-1}+v\cos\left(\theta_{k}\right)dt\\ y_{k}=y_{k-1}+v\sin\left(\theta_{k}\right)dt \end{array}\right.$$

Although this model does not accurately reflect actual vehicle behavior, the small available steering angle limits the impact of this approximation. Though our control space is of type (v,δ) instead of (v,w) in the original approach, the principle remains the same. As the maximum admissible steering angle on the vehicle is $18^{\circ }$ in automatic mode and as it has two steering axles, we control it by applying symmetrical steering. This means calculating the steering angle corresponding to a vehicle half the length of the EZ10, then applying this angle to the front wheels and its opposite to the rear.

### 4.2. Path Following

Our first experiments were aimed at validating the proposed path-following module without obstacles on the track. We compared it to a traditional and easy-to-implement control law, namely, Pure Pursuit [14], which was used in a number of vehicles during the 2005 DARPA Grand Challenge [33]. Our goal here was to verify whether our approach could succeed in classical path-following before using it for obstacle avoidance.

In this scenario, the reference trajectory was obstacle-free and the vehicle was required to complete a full circuit starting from a specific point.

We discretize the available control space, the dynamic window, into five steering angles and keep the longitudinal speed constant (0.8 m/s) for this scenario. The predictions are for a 5 s temporal horizon.

We observed that both techniques generally followed the reference trajectory on straight sections (Figure 8). However, there were significant deviations on bends, particularly at the roundabout exit. Pure Pursuit exhibits slightly better path tracking performance over a significant portion of the trajectory. However, it does not take into account the capabilities of the actuators, which leads to strong commands sent by the controller at the beginning of the trajectory and during the exit of the roundabout to quickly join the trajectory, resulting in vehicle oscillation and large lateral deviation errors (see Figure 9).

Table 2 allows these results to be compared numerically. We set a maximum tolerable lateral deviation threshold of 50cm from the reference trajectory, which indicates the required tracking accuracy in our application. We observed that the Pure Pursuit control law was able to keep the robot within the acceptable range 91% of the time, while with CSD it stayed within this range 97% of the time.

According to our application, the errors generated by our strategy are acceptable. Our approach, which takes into account the vehicle’s dynamics, allows for smoother control and ensures passenger comfort by avoiding abrupt steering maneuvers (Figure 10). All of these results validate the path-following module.

### 4.3. Path-Following and Obstacle Avoidance

After validating the path-following approach, we integrated the obstacle avoidance module. The entire configuration of the system and the connection between the different blocks is summarized in the diagram shown in Figure 11.

In this stage, two obstacles were placed on the track. We placed sidewalk-side cones near the obstacles to force avoidance on the left side and because the road edges were not high enough to be detected by our lidars.

The discretization of the steering space is maintained with five admissible steering angles. The controller is enabled to select and calculate an additional speed in the dynamic window (lower than 0.8 m/s) to allow the vehicle to slow down if necessary. Admitting more speeds and even more steering angles depends on the available computational capabilities.

At each iteration, the controller tests two speeds v × five steering angle δ, i.e., ten candidate trajectories. The vehicle has a control frequency constraint set at 10 Hz, i.e., a control couple (v, δ) must be transmitted at least ten times in one second. The whole operation of selecting the best trajectory is then carried out in 0.1 s at most.

Two experiments were conducted under this configuration. In the first scenario, we used navigation with a good localization provided by the vehicle’s Kalman filter, which combines odometry, GPS measurement, and/or visual odometry.

The second scenario involved injecting uncertainty into the vehicle’s localization to test its obstacle avoidance capabilities under these conditions. In practice, we want to simulate errors in localization that will influence path-following in order to observe the impact on obstacle avoidance. To enforce these effects, it is necessary to bias the mean of robot estimated state. This is crucial because as long as the mean stays close to the ground truth, the path-following will not be affected regardless of the covariance of the position. At this stage, it is important to clarify that EZ10-type vehicles are disproportionately large compared with the experimental vehicles for which PAVIN was designed. Thus, for safety reasons, as our lidars were not able to detect road edges and in light of the narrowness of the roads in PAVIN, we injected a bias of 50 cm in the x and y directions on the output of the Kalman filter.

The results of both scenarios are shown in Figure 12.

Figure 13 depicts the obstacle avoidance maneuver for Obstacle 1 in the first scenario through successive images.

In both scenarios, the vehicle avoided the obstacles with the optimum (smallest possible) distance by passing close to them.

In the second scenarios, the bias injected into localization logically resulted in offset and degraded path-following. When passing close to the first obstacle, the vehicle was shifted to the right on the road due to biased localization, making the avoidance closer than when it was localized and following the path in the middle of the road.

Furthermore, in the scenario without uncertainty we observed that path-following in the bends was better due to the addition of speed control, which allowed the vehicle to slow down while turning.

On passing the obstacle on the left side, the measurements from the right lidar (Figure 14) indicate that the vehicle passes at approximately 20 cm from the obstacles in both scenarios; this value is quite logical, as the grid cell size is 20 cm. These experiments demonstrate that obstacle avoidance is agnostic to localization errors in our approach, as the distance between the vehicle and obstacles is as small as possible regardless of the accuracy of the vehicle’s position in the world.

## 5. Conclusions

In this work, we propose an approach that combines trajectory following and obstacle avoidance while managing the impact of localization uncertainty on both objectives. While path following is affected by localization inaccuracy, we demonstrate that obstacle avoidance remains optimal due to the fact that the vehicle’s local perception is not related to its absolute localization. We carried out real tests with an EZ10 shuttle on the PAVIN experimental platform, and the results validate the proposed concept.

The purpose of this technique is to enable navigation in rural areas and environments where precise localization is not guaranteed.

The main limitation of this strategy is the strong assumption made about orientation, which we assume to be unaffected by uncertainty. As the map is robot-centered and does not rotate with the vehicle, uncertainty about orientation would distort the data on this map. One solution would be either to embed a map that rotates with the vehicle, which is computationally expensive, or to equip the vehicle with cameras to extract the vehicle’s orientation accurately and instantaneously using visual odometry.

Future perspectives include:Additional tests modeling and injecting more realistic noise or localization errors, such as GPS signal loss, to observe robustness against uncertainty.Integration of a prior map into the local map, for example, to take into account road boundaries, which were not detectable by the planar lidars used in this study.Translating path-following into a risk map that can be associated with the occupancy grid in order to define more unifiable criteria.

## Figures and Tables

**Figure 1 sensors-23-07237-f001:**
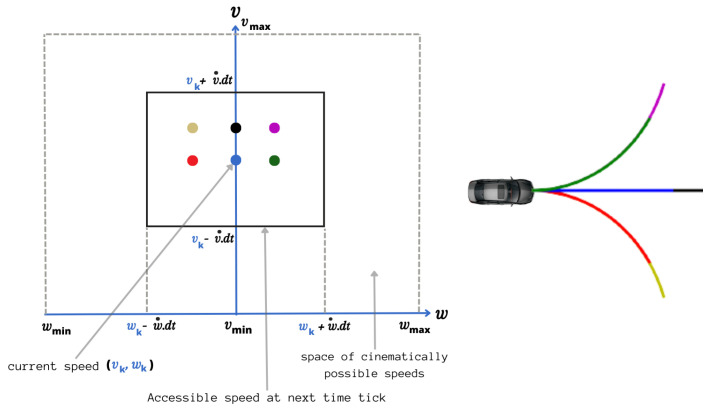
Command selection space taking into account robot actuator dynamics.

**Figure 2 sensors-23-07237-f002:**
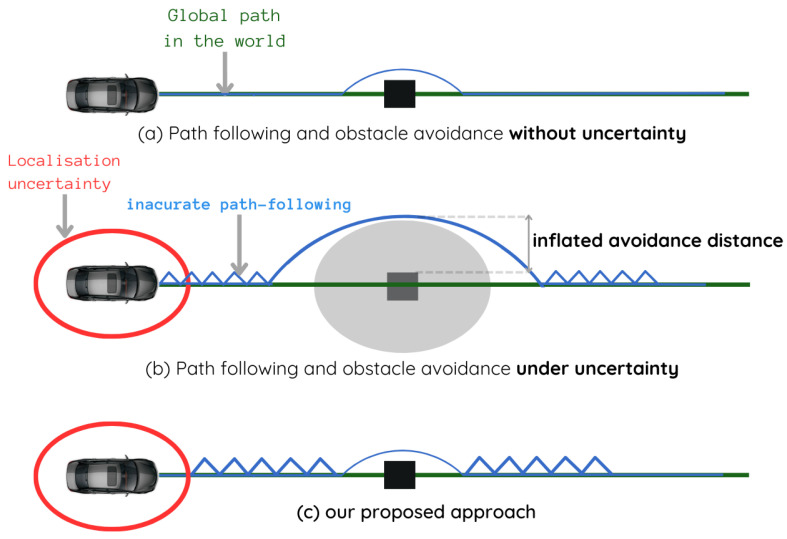
Uncertainty effect on navigation: (**a**) navigation with accurate localization; (**b**) common approaches to uncertain localization; (**c**) our proposed approach to uncertain localization.

**Figure 3 sensors-23-07237-f003:**
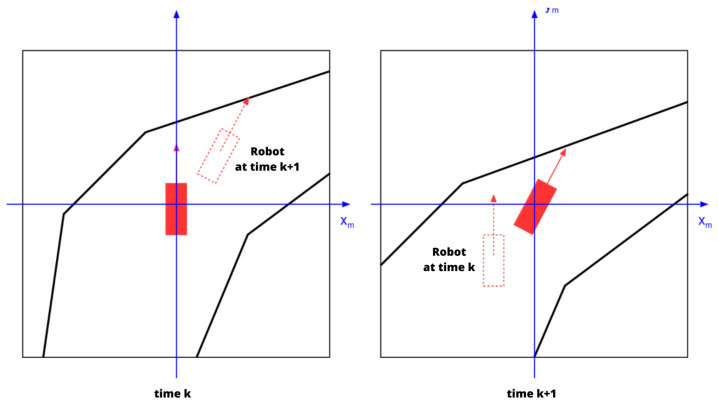
Robot’s movement on the local map. The map “slides” under the robot in such a way that the robot maintains its central position; the map itself does not rotate.

**Figure 4 sensors-23-07237-f004:**
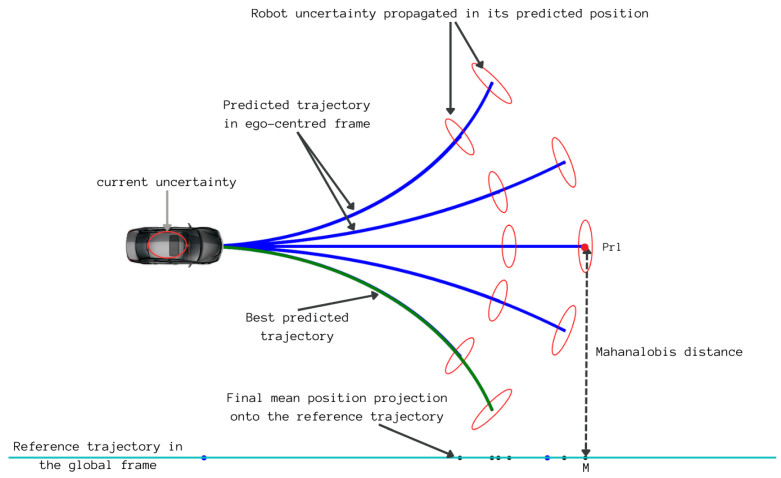
Uncertainty propagation over the predicted trajectory of the robot.

**Figure 5 sensors-23-07237-f005:**
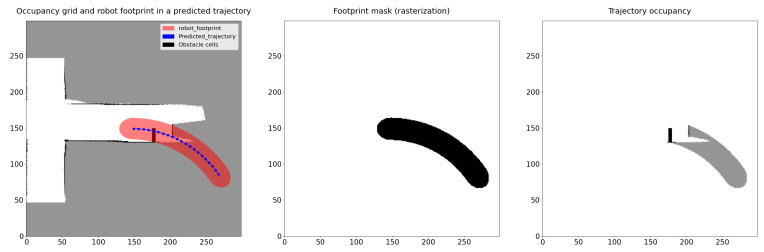
Path occupancy probability extraction.

**Figure 6 sensors-23-07237-f006:**
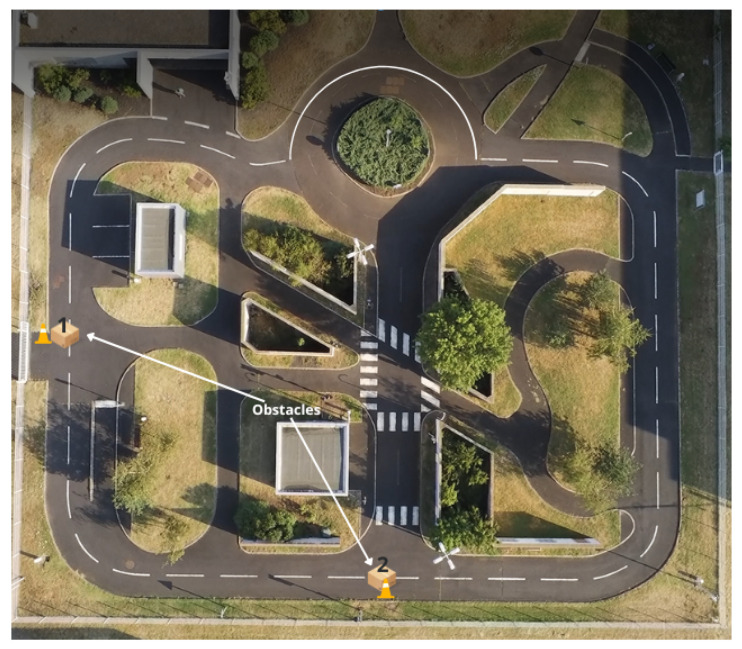
PAVIN: Our reference trajectory is the large loop described by the dashed curve passing through the solid part of the roundabout. Obstacles (1 and 2) are placed at certain locations that are not specified to the controller.

**Figure 7 sensors-23-07237-f007:**
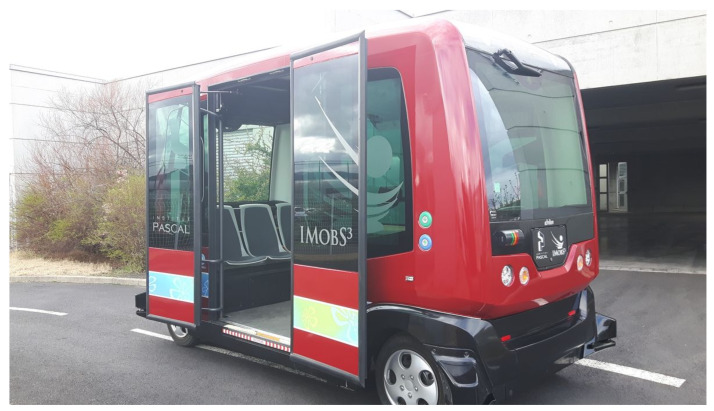
The EZ10 shuttle vehicle used for our experiments.

**Figure 8 sensors-23-07237-f008:**
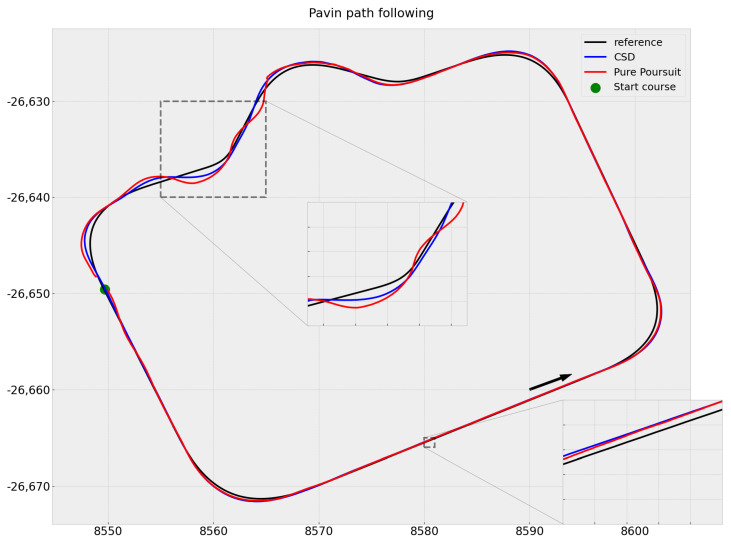
Path-following: reference path in black, Command Space Discretization (CSD, our approach) in blue, and Pure Pursuit in red.

**Figure 9 sensors-23-07237-f009:**
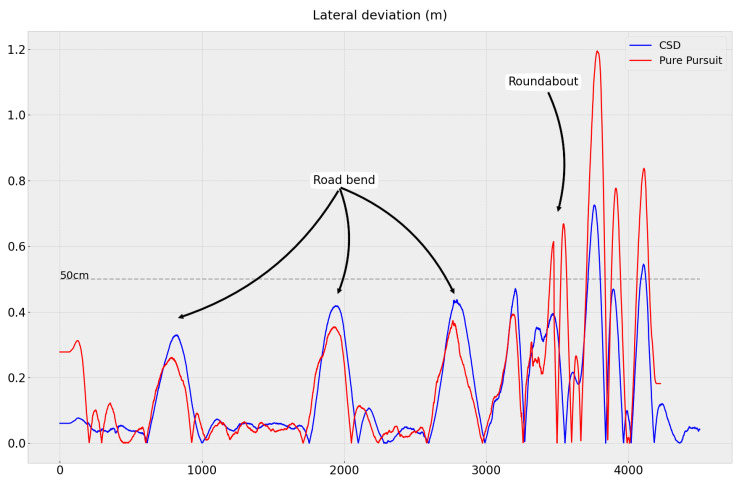
Vehicle lateral deviation comparing the Pure Pursuit algorithm and proposed CSD approach.

**Figure 10 sensors-23-07237-f010:**
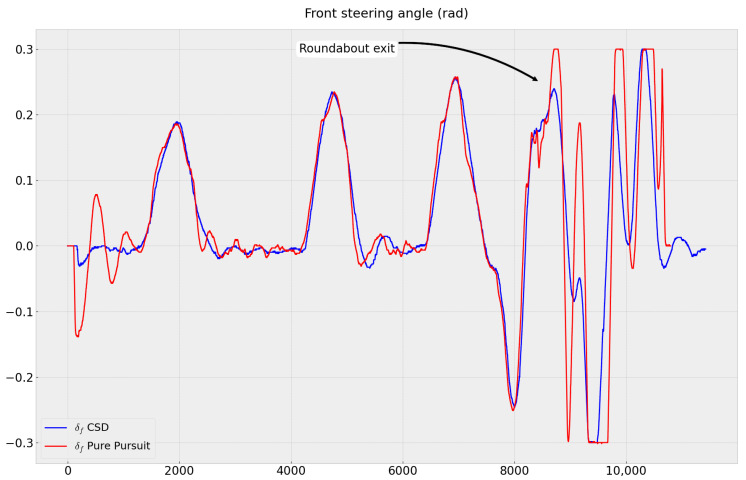
Front wheel steering angles sent to vehicle.

**Figure 11 sensors-23-07237-f011:**
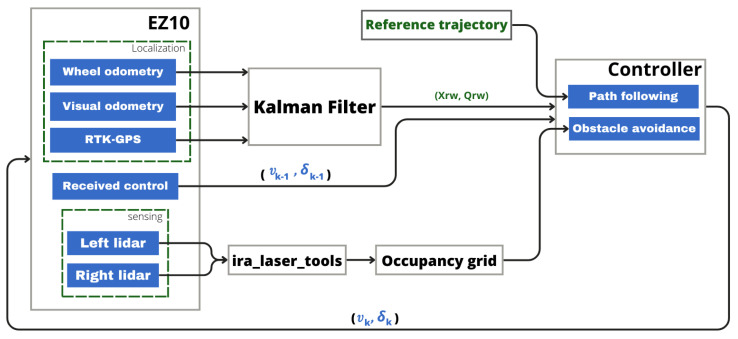
System block diagram.

**Figure 12 sensors-23-07237-f012:**
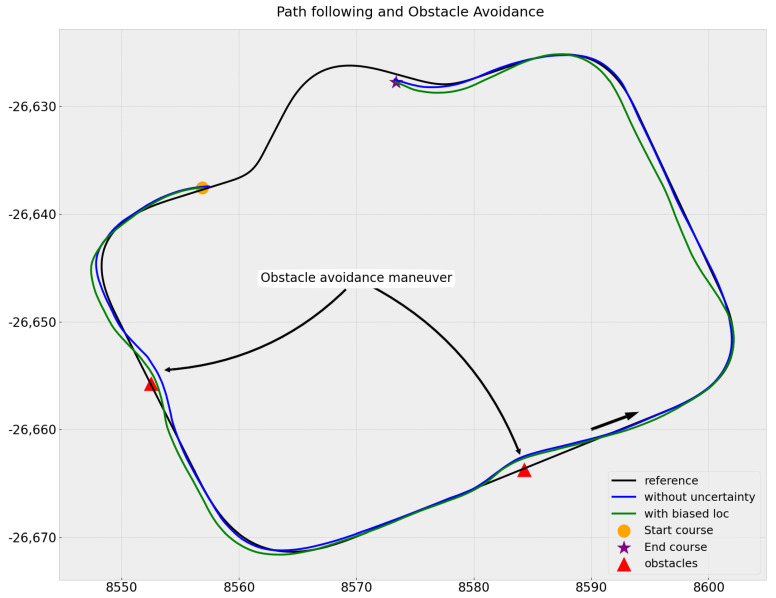
Path-following and obstacle avoidance, showing the scenario without uncertainty in blue and the scenario with biased localization in green.

**Figure 13 sensors-23-07237-f013:**
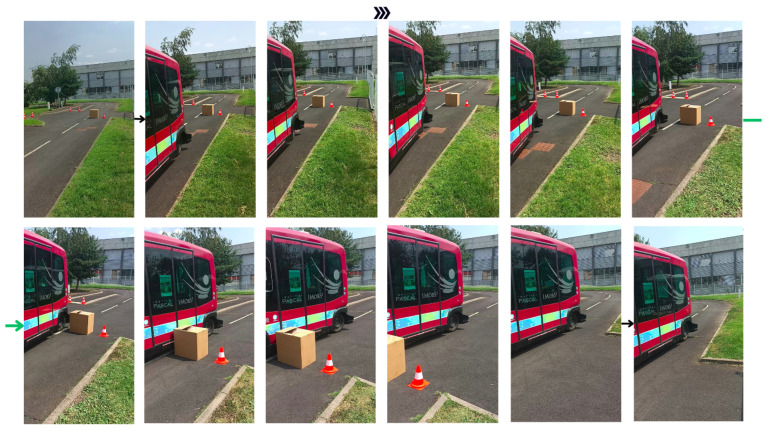
Obstacle avoidance maneuver; the images follow each other from left to right and continue on the second row.

**Figure 14 sensors-23-07237-f014:**
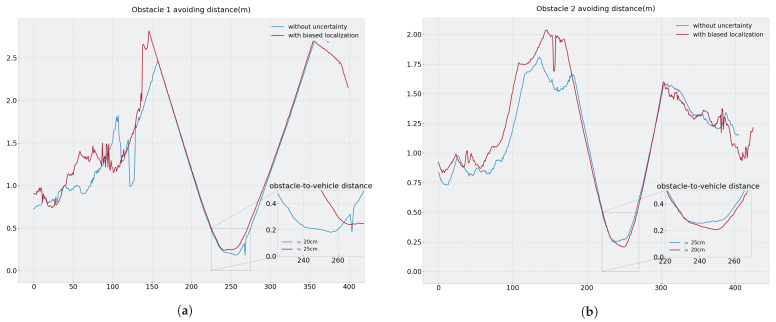
Right lidar minimal scan range ρ during obstacle avoiding maneuver in both scenarios: (**a**) Obstacle 1 and (**b**) Obstacle 2.

**Table 1 sensors-23-07237-t001:** EZ10 technical specifications.

Characteristics	Values
Dimensions (L × W × H)	4.050 × 1.892 × 2.871 (m)
Wheelbase (L)	2.80 m
Maximum steering $\delta_{max}$	0.3 rad
Maximum steering speed $\omega_{max}$	0.2 rad/s
Maximum speed $v_{max}$	11 m/s (40 km/h)
Maximum acceleration $a_{max}$	0.5 m/s²
Minimum control frequency	10 Hz
No-load mass	1700 kg
Maximum mass	2800 kg

**Table 2 sensors-23-07237-t002:** Comparative table of tracking errors.

Approaches	Mean	Std	≤50 cm
Pure Poursuit	0.19 cm	0.21 cm	91.65%
CSD	0.15 cm	0.15 cm	97.34%

## Data Availability

Some or all data, models, or code that support the findings of this study are available from the corresponding author upon reasonable request.

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
