# Peer review of "A Framework for Optimal Navigation in Situations of Localization Uncertainty"

_sensors, 2023, doi:10.3390/s23167237_

Round 1

Reviewer 1 Report

This manuscript presents a framework of navigation with localization uncertainty for AVs. The method is given in an acceptable way and the results of the experiments are well shown. I suggest the authors to consider the following perspectives:

Is the size of the obstacle relevant to the performance? Besides, it's better to have a photo of the obstacle on the road, to show the environment. 

Would there be cumulative uncertainty, especially when there are many obstacles close to each other?

In general, it would be interesting to show its performance in more situations.

In general, the language is easy to understand. Some of the sentences need  to be simplified/edited, e.g. L325-327.

Reviewer 2 Report

n the "Sense-Plan-Act" strategy it should be referenced.

In many studies change to studies as it only references two citations unless it says "in studies reviewed or found".

multiple sensor data== perhaps say data from different sources. 

There are abbreviations that do not indicate meaning or equivalence.

there is a sentence that says "Using this framework, not only does uncertainty 150 affect the robot's global objectives", it is not clear what the framework is, what they are proposing, or which framework they are referring to?

There are errors in the English writing 

They should describe more the applied experiment, what was the task that the EZ10 performed, or the route (Figure 6). They show it but it is not described in detail, how to get from point A to point B, where there would be a number of obstacles and describe why they are not specific to the controller...
